# Physical Activity Participation and the Environment in Children and Adolescents: A Systematic Review and Meta-Analysis Protocol

**DOI:** 10.3390/ijerph18126187

**Published:** 2021-06-08

**Authors:** Longxi Li, Michelle E. Moosbrugger, Yang Liu

**Affiliations:** 1Department of Physical Education and Health Education, Springfield College, Springfield, MA 01109, USA; lli@springfieldcollege.edu; 2Physical Education College, Hebei Normal University, Shijiazhuang 050024, China; liuyang1982@hebtu.edu.cn; 3Provincial Key Lab of Measurement and Evaluation in Human Movement and Bio-Information, Hebei Normal University, Shijiazhuang 050024, China

**Keywords:** children, adolescent, ecological frameworks, physical activity, health

## Abstract

Physical activity (PA) and sports are efficient ways to promote the younger generation’s health and wellbeing. However, evidence is limited due to heterogeneous samples and measurements. This study aims to identify promoting and inhibiting correlates associated with children’s and adolescents’ non-organized PA participation and further demonstrate the complexity of PA and ecological factors. A systematic review and meta-analysis will be applied by following the preferred reporting items for systematic review and meta-analysis protocols (PRISMA-P). Seven bibliographic databases (PubMed, SPORTDiscus, PsycInfo, MEDLINE Complete, ERIC, Dimensions, and Academic Search Complete) will be systematically searched to identify eligible articles based on a series of inclusion and exclusion criteria. Inclusion criteria are that the study: (a) is not classified as a systematic review with or without meta-analysis; (b) is published in last 20 years; (c) includes children and adolescents; (d) quantitively measures PA; (e) includes review of ecological factors. The internal validity will be evaluated using a validated quality instrument. Calculations will be produced in SPSS 27.0 and Comprehensive Meta-Analysis 3.3. This study will provide evidence and address the questions regarding the factors that significantly impact children’s PA participation and limitations regarding the design, sampling, and measurement in currently selected studies. PROSPERO registration number: CRD42021244918.

## 1. Introduction

Daily active play and physical activity (PA) have traditionally been an essential part of life for children and adolescents, and are highly recommended due to the associated health and environmental benefits [1,2,3,4]. However, less than 20% of children worldwide meet the recommended PA guidelines [5]. Computers and social media have increased sedentary time and further decreased the need and desire for children to move and play [6,7]. Moreover, participation in PA decreases with age [8,9,10], and the decline is more significant in girls than boys [11]. In light of these statistics, increasing the knowledge about what types of health promotion interventions are most effective for improving children and adolescent health behaviors is critical for policymakers and interventionists working to improve health outcomes [12]. The challenges associated with getting children and adolescent groups active every day should be met with age-appropriate PA, enthusiastic leadership, and family and friendly support [13].

Childhood and adolescence represent important developmental periods where lifelong habits can be established to promote healthy lifestyles and reduce chronic disease risk in early adulthood [14]. Frequently participating in PA brings positive impacts on health and fitness, especially in children and adolescents [15,16,17,18,19]. Meanwhile, children and adolescents who do not participate in adequate PA are more likely to engage in smoking, increased screen time, and poor dietary habits. Developing this pattern of behaviors during childhood and adolescence enhances the risk for continuing these patterns and experiencing poor health in adulthood [20,21]. Despite the potential of early efforts, children and adolescents present unique challenges for health promotion programs because their behavior is significantly influenced by multiple ecological systems and individual factors, such as developmental age, family, neighborhoods, cultural context, school systems, and policies that govern these systems [22,23,24]. 

The primary influential factors on sports and PA participation during childhood and adolescence are tightly related to their surrounding environment, which has commonly been examined through ecological frameworks [14,16,25]. Through the past several decades, the ecological frameworks have been developed in the literature [26,27,28]. Stokols [29] noted works contemporaneous with Bronfenbrenner were influential in developing socioecological approaches to health promotion (see Figure 1). 

McLeroy and colleagues [33] proposed an ecological model that emphasized the role of both social and environmental factors reflecting Bronfenbrenner’s [30] four levels, which primarily focused on an interactive and cumulative effect on change. The construct of theory found by Bronfenbrenner was broadly conceptualized in the Socioecological Model (SEM) and concentrated on the principal contributors that might affect health. The SEM essentially argued that health and wellbeing were influenced by the interaction among the characteristics of the individual, the community, and the environment [38]. From a developmental perspective, it made intuitive sense that interactions at each of these levels could affect change in a specified sociocultural context for the individual. The Centers for Disease Control and Prevention (CDC) adopted the SEM for numerous health promotion endeavors in order to comprehensively consider the integration effects of interpersonal, organizational, community, and policy factors on a specific population [26]. 

Furthermore, the SEM provides a theoretical framework to examine multiple factors influencing PA behavior [36,39]. The basis of the SEM is the assumption that the combination of individual, social–environmental, and physical environmental factors will best explain PA participation in children and adolescents [40]. Along this line of consideration, the systems approaches (SA) and youth physical activity promotion (YPAP) models were newly added to specify SEM, which mainly focused on the correlates of PA behavior change in youth, specifically in school-age (elementary, middle, and secondary) children and adolescents [36]. Both succeeding models were rooted in SEM and Bronfenbrenner’s [30] ecological framework, where personal, social, and physical environmental factors might influence, strengthen, or empower youth to be physically active [36]. Although SA and YPAP models provided an attractive conceptual basis for the ecological framework, there was limited theoretical and practical evidence of those models in examining PA participation in children and adolescents [40,41].

In light of the variety of ecological frameworks, this study will align with Bronfenbrenner’s [30] ecological framework and its branching models. This study will provide a rigorous protocol in facilitating future researchers to screen previous literature and systematically reviewing the influencing factors that align with the ecological frameworks. This perspective emphasized the broad range of situations and contexts individuals may encounter, including the aforementioned systems and factors that directly and indirectly affect and modify an individual’s biological development [24,42,43]. In this study, the evidence will be extracted and categorized into five broad categories regarding individual, micro (family, peers, neighborhood, and school), exo (extended family), macro (country level physical environments, policy, and sociocultural context), and chrono (considering timeline of change). With the expectation of broadening the scope of knowledge, we expect to examine variability in effect sizes using study-level variables and then explore the relative impact of influence factors within ecological systems. Hence, this systematic review and meta-analysis protocol will aid in answering the following research questions: (1) What are the promoting/inhibiting factors that significantly impact children’s PA participation, and which is the most influential factor for child and adolescent PA within the ecological system? (2) What are the limitations regarding the design, sampling, and measurements in current selected studies? (3) What is the current trend and future direction of this research area?

## 2. Materials and Methods

This review protocol has been registered in the PROSPERO international prospective register of systematic reviews; the registration number: CRD42021244918. The Preferred Reporting Items for Systematic Review and Meta-analysis Protocols (PRISMA-P) will serve as guidelines for reporting present protocol and subsequent formal meta-analysis. Electronic databases served as the primary method of identifying eligible studies. Manual searches of relevant papers will be also carried out. The current study will also conduct backward reference searches of other reviews and consult with experts in physical education and health education.

### 2.1. Search Strategy

The search strategy included use of the following databases: PubMed, SPORTDiscus, PsycInfo, MEDLINE Complete, (Educational Resources Information Center) ERIC, Dimensions, and Academic Search Complete. The search strategy will align with the guidance of the Cochrane handbook. Keyword combinations employed in the search are listed in Table 1.

### 2.2. Inclusion and Exclusion Criteria

#### 2.2.1. Study Designs

We will only include original empirical studies, which should not be classified as a systematic review with or without meta-analysis, published between 2001 and 2021. Type of studies include two categories, randomized controlled trials (RCT) and observational studies (non-randomized studies, or NRS). More specifically, we will include cohort, case control, interrupted time series, cross sectional, case series, case report, and other types of observational studies. Furthermore, only quantitative data will be extracted if any mixed-method studies meet the inclusion criteria. 

#### 2.2.2. Participant

Participants will include those that are 3–18 years old based on the definition of the time period of childhood and adolescence by the US Department of Health and the Food and Drug Administration (FDA) and Bright Futures (American Academy of Pediatrics). Subjects older than 18 years at the start of the study will be excluded, as will children and adolescents who are affected by a particular disease, disorder, injury, or trauma at the time of the intervention. Considering that the longitudinal study may meet the inclusion criteria, we will include the study if participants are ages 3–18 years at the start of the study, even if they turn 19 during the course of the study.

#### 2.2.3. Intervention

We will accept all types of influential factors regarding Bronfenbrenner’s [30] ecological framework and socioecological models (see Figure 2). The ecological factors include individual characteristics, family, peers, school, community, social context, policy, and weather within ecological systems [14,30,31,32,33,34,35,36,37]. In selected studies, participants should be exposed within the multiple-level ecological factors in a manner that influences the targeted population’s PA participation. 

#### 2.2.4. Outcome Measures

According to the World Health Organization (WHO), the definition of PA is, “any bodily movement produced by skeletal muscles that require energy expenditure” [4]. In this study, all measurement results of PA within selected studies will be considered as outcomes. The effects of ecological factors on children’s and adolescents’ PA participation will be primarily evaluated by the changes between pre- and post-intervention or the designated time period, or differences across age groups in a cross-sectional study. Aligning with the expectation of this study, we will extract quantitative outcome measures of PA in selected studies (e.g., accelerometers and heart rate data). More specifically, the inclusion study has as its outcomes any measure of overall or general PA, light PA, vigorous PA, moderate PA, and moderate to vigorous PA in frequency, intensity, or duration. Self-reported PA is correlated with accelerometer data, and these correlations tend to be moderate to high [44,45]. Hence, children and adolescent PA participation data will be accepted in either subjective or objective format. In this way, objectively assessed PA will be examined in aggregation with self-reported PA. Therefore, secondary outcomes will mainly involve adverse events, parent-reported, and self-reported outcomes.

### 2.3. Studies Selection and Data Extraction

During the initial screening stage, two authors will select the title and abstract of the literature. Recording and managing related literature will be accomplished with Mendeley software (Mendeley Ltd, London, UK). Next, two authors will review full texts of selected studies with the inclusion criteria. Finally, two authors will synthesize their screening results and determine the preliminary study pool. If a study meets the inclusion criteria but does not report results in the suggested variables, we will remove the study in the validation process because study outcome(s) should include relative variables. Accordingly, we will narratively review the excluded study’s method and conclusion in our perspective paper. The project administrator will serve as the coordinator, dealing with potential disagreements. In this way, internal discussions and then consultation with an experienced researcher in public health and physical education are considered two solutions for disagreements. The aforementioned selection and screening process will be depicted in the preferred reporting items for systematic review and meta-analysis (PRISMA-P) flowchart (see Figure 2).

In the selecting process, included study will accord with the PRISMA-P statement. The authors will remove duplicate studies from different databases by evaluating titles and abstracts, and then evaluate full texts of the eligible studies. Finally, the authors will conduct qualitative and quantitative evaluations on inclusion studies [46].

### 2.4. Assessment of Risk of Bias

The Cochrane Collaboration’s tools will be applied to assess confounding, selection of participation, classification of interventions, deviations from interventions, missing data, measurement of outcomes, and selection of the reported results in selected RCT and NRS [47]. Each of these seven domains will be examined across all included studies to determine whether the risk of introduced bias was low, moderate, or high by two independent reviewers. We will assess the strength of recommendations with the GRADE system for studies. Considering both RCT and NRS might be included, we will use GRADE system but the risk of bias in non-randomized studies of interventions (ROBINS-I) tool as part of GRADE’s certainty rating process will be used for observational/non-randomized trials [48]. The use of GRADE with ROBINS-I allows for two independent reviewers to assess the risk of bias in RCTs and non-randomized studies on a common metric. The gradings of evidence quality and risk of bias reporting by two reviewers will be compared, and the third reviewer will be involved if any disagreements occur. The results of the grading of evidence quality and risk of bias will then be generated as a risk table [49].

### 2.5. Data Analysis and Synthesis

#### 2.5.1. Data Analysis Procedure

To the best of our knowledge, there are sufficient studies published in last two decades, so the likelihood of a meta-analysis is well-supported. In the follow-up meta-analysis, the review management software Comprehensive Meta-Analysis (version 3.3; Biostat, Englewood, NJ, USA) and SPSS (version 27.0; IBM Corp., Armonk, NY, USA) will be applied. The meta-analysis will include four procedures: first, for data entry, obtain correlations between PA and ecological factors along with sampling error variances; second, for data screening, screen for outliers, defined as correlations whose residuals had z-scores either greater than 3.29 or less than −3.29: a z-score of ±3.29 cut off 0.1% of scores with 99.9% of z-scores lying between −3.29 and +3.29, the threshold of normal distribution in this protocol [50]; third, estimate overall effects and heterogeneity; finally, perform publication-bias analyses. To ensure appropriate weighting of each individual study in the meta-analysis, studies contributing multiple outcomes will be aggregated separately such that each one contributed a single effect size to the analysis [44]. As treating non-independent studies as independent, this approach will result in more accurate standard errors and reduce biases [51,52].

#### 2.5.2. Data Synthesis

Furthermore, we will categorize all articles and extract data from the final sample. In studies that reported a correlation between two continuous variables, the Pearson’s correlation coefficient *r* could serve as the effect size index [50]. In this way, Pearson’s *r*, means, standard deviations (SDs), sample size, F-tests, *t*-statistics, *p* values, and effect size (odds ratios) will be extracted. If these are not applicable, other relative statistics will be extracted, to calculate effect sizes (summary correlation ′*r*), 95% confidence intervals, and standard errors (SEs) of the corelates. All effect sizes will be converted to Fishers’ Z and then summary ′*r* to allow for comparisons across studies. Specifically, we will use the following procedure to generate the effect size. First, input extracted data into Comprehensive Meta-analysis software; second, the program will calculate correlations to Fisher’s Z transformation and perform the analysis using this index; finally, Fisher’s Z will be conversed back to summary correlation ′*r* as the effect size [51,53,54]. The guidelines of Warner [50] will be applied to interpret effect sizes *r* by assigning qualitative descriptors as follows: 0–0.3 as small, 0.3–0.6 as moderate, and 0.6 and above as large.

#### 2.5.3. Heterogeneity and Publication Bias

The heterogeneity test between studies will be assessed regarding the Q and I^2^ statistics. Applying the recommendation of Cushing et al. [44], a fixed-effect model will be conducted in the analysis if *p* > 0.01, I^2^ < 50%. Otherwise, a random effect model will be applied due to the high level of heterogeneity if *p* < 0.10, I^2^ > 50% [44]. In addition, sensitivity analysis will be utilized to further reduce heterogeneity by removing studies with a high risk of bias. Meta-regression on covariates in terms of age, gender, and quantitively measured PA participation (e.g., duration, frequency, and intensity) will be used to assess sources of heterogeneity. Finally, funnel plots and the Egger test will be employed to evaluate the publication bias.

### 2.6. Subgroup Analysis

Subgroup analyses will be performed on individual characteristics (e.g., gender, age, and ethnicity) and ecological systems (e.g., microsystem, exosystem, macrosystem). The purpose of subgroup analyses is to specify the effectiveness of ecological systems and factors in influencing non-organized PA participation in children and adolescents.

### 2.7. Data Visualization

The researchers will apply the co-occurrence analysis to demonstrate the similarities and differences within the selected studies via VOSviewer (version 1.6.16; Centre for Science and Technology Studies, Leiden, The Netherlands [55]). This procedure aims to identify “hot spots” of themes and trends in publications of PA, public health, children and adolescents, and environmental research. The co-occurrence analysis will be calculated on a co-occurrence matrix, more specifically, the construction of a map will follow a three-step process. Firstly, a similarity matrix will be calculated based on the co-occurrence matrix; Accordingly, a map will be constructed by applying the VOS mapping technique to the similarity matrix, weighted on total link strength; Finally, the map will be translated to illustrate the themes and trends among studies [55].

### 2.8. Ethics

This study will be based on findings of previous studies so that ethics approval is not required. Further, the results will be submitted to a peer-reviewed journal of the relative field for consideration of publication.

## 3. Expected Results/Discussion

Following a year of social restriction because of COVID-19, motivating children and adolescents back to physical and social activity should be considered a priority in the coming post-pandemic era. Completing this protocol will provide evidence regarding child and adolescent non-organized or spontaneous PA participation and engagement. Furthermore, we believe that this study may lead to several recommendations for caregivers, physical educators, policymakers, and pediatric researchers, such as elucidating the influence factors that are significantly correlated with child and adolescent PA participation; how effective they are; what different effects result from those factors acting on child and adolescent PA participation, especially across different genders and ethnicities.

## 4. Conclusions

Predicting PA during childhood and adolescence is a complex topic. To the best of our knowledge, this will be the first comprehensive systematic review and meta-analysis to shed light on the effects of ecological and socioecological factors on child and adolescent PA participation. The current study will be key to providing evidence for future research in examining child and adolescent non-organized or spontaneous PA participation and multiple layers of ecological factors. This will also aid agencies and organizations in effectively motivating children and adolescents back onto the playground and to remain physically active, which should be considered as the priority in the post-pandemic era.

## Figures and Tables

**Figure 1 ijerph-18-06187-f001:**
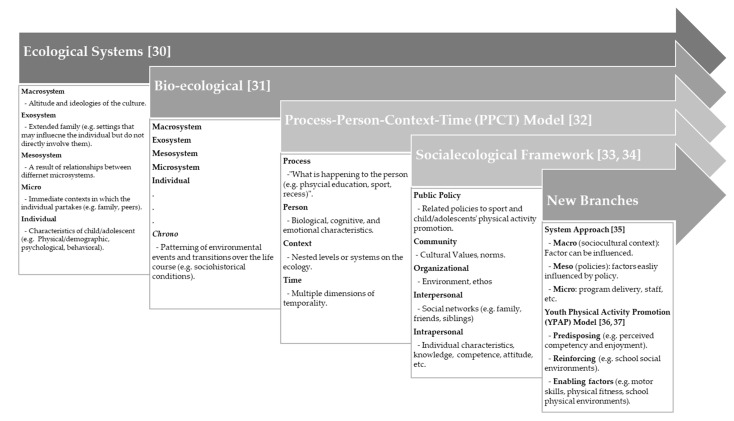
Diagram of the theoretical frameworks regarding child and adolescent development and physical activity participation [30,31,32,33,34,35,36,37].

**Figure 2 ijerph-18-06187-f002:**
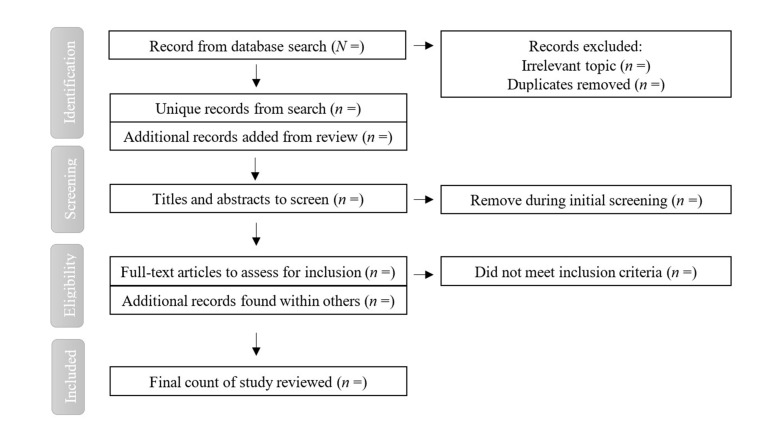
PRISMA flowchart of searching and screening.

**Table 1 ijerph-18-06187-t001:** Search Strategy.

Concept	Keywords
I. Outcome: PA participation and engagement	^1,3^participation or ^2^physically active or ^1,4^leisure activity [MeSH Terms] or ^1,4^execrise [MeSH Terms] or ^1−7^physical activity or ^1−7^vigorous physical activity or ^1−4,7^moderate to vigorous physical activity or ^1−4,7^light physical activity or ^1−4^sedentary behaviors or ^1,4^BMI [MeSH Terms] or ^1−4,7^self-efficacy.
II. Participants: Children and Adolescent (3–18-years-old)	^1–7^youth or ^1–7^adolescent* or ^1−6^young people or ^1−6^ teen or ^1−6^young adult* or ^1−6^children or ^1−6^kid* or ^1−7^teenager*. Exclude (NOT): ^1−7^smoking or ^1−7^drinking or criminal* or ^1−7^disability or ^1−7^injury or ^1−7^handicap or ^1−7^disorder or ^1−7^resistance training or ^1−7^injuries or ^1−7^accident or ^1−7^trauma or ^1−7^older adult* or ^1−7^elderly or ^1−7^seniors or ^1−7^geriatric*.
III. Exposure: Influence factors	^1−7^Family or ^1−7^peer* or ^1−7^community or ^1−7^school or ^1−7^coach or ^1−7^teacher or ^1−7^social context or ^1−7^social-economic or ^1−7^socioeconomic status or ^1−7^friend* or ^1−7^individual* or ^1−7^policy or ^1−7^cultur* or ^1−7^health promotion interventions or ^1–7^physical education session*.
IV. Exposure: Exercise types	^1−7^physical exercise or ^1−7^exercise* or ^1−7^fitness or ^1−7^physical activity or ^1−7^sport* or ^1−7^walking or ^1−7^non-organization sport* or ^1−7^recreational sport* or ^1−7^motor activity or ^1−7^leisure activity.
V. Theoretical Framework	^1−7^Ecological theory or ^1−7^ecological framework or ^1−4^PPCT or ^1−7^social ecological or ^1−7^social ecological theory or ^1−7^socioecological theory or ^1−7^socioecological framework or ^1−7^socioecological.
VI. Design	^1−7^experimental study or ^1−7^experimental research or ^1−7^quasi experimental study or ^1−7^empirical study or ^1−7^quantitative study or ^1−7^longitudinal study or ^1−7^cross section* or ^1−7^mixed-method or ^1−7^social network analysis or ^1−7^random control trail or ^1−7^cluster random control trail or ^1,4^RCT or ^1,4^clinical trial or ^1,4^treatment outcome.

^1^ PubMed, ^2^ SPORTDiscus, ^3^ APA PsycInfo, ^4^ MEDLINE Complete, ^5^ ERIC, ^6^ Academic Search Complete, and ^7^ Dimensions; range of publishing is from 2001 to 2021. The * is used for truncation within database searches to allow a search for the root of a word.

## Data Availability

Not applicable.

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
