# Peer review of "Physical Activity Participation and the Environment in Children and Adolescents: A Systematic Review and Meta-Analysis Protocol"

_ijerph, 2021, doi:10.3390/ijerph18126187_

Round 1
Reviewer 1 Report
This study protocol describes a review/meta-analysis approach that will include multiple factors, a wide age range, and various theories. It is a much-needed meta-analysis I think, and therefore I appreciate this work a lot.
I still have some minor comments.
Introduction:
P.1, L.31, Please properly introduce abbreviations (PA) before using them
P.1, L.41, replace physical activity with PA
P.2, L.45-48, please rephrase these two sentences as they do not read well, e.g. adjust the phrases to have the same subject
P.2, L.71, replace this with it
Method:
P.3, L.101-105, please insert (1) and (2) for the research questions, it reads better
P.6, L. 193, please state on what it depends if you can do a meta-analysis
P.6, L. 197, please state why a z-score of 3.29 or greater will be handled as an outlier (even though you cited this, to make it clear to the reader please state explicitly the reason why it can be handled this way
Discussion:
P. 7, L. 255-259, change from questions to statements since these are not research questions but expected results
Author Response
Thank you for your kind words and feedback! We have addressed your comments as follows:
- P.1, L.31 - We added the introduction of the abbreviation for physical activity (PA) in both the abstract and in the introduction on p. 1. The remainder of the document was reviewed to ensure other abbreviations were introduced prior to being used. The introduction of abbreviations for light physical activity, moderate physical activity, and moderate to vigorous physical activity were removed because the abbreviations were not subsequently used.
- P.1 L.41 - "Physical activity" was replaced with "PA" on here as well as on p. 5 lines 164-168, and multiple places on pp. 5 and 6.
- P.2 L.47-49 - was revised for clarity.
- P.2, L.71 - "this" was replaced with "it".
- P.3, L.103-106 - numbering was added for the research questions.
- P.6, L. 218-220 - the meta-analysis will be conducted if adequate studies are identified. We have confidence that meta-analysis will be applied because there are adequate studies that are sufficiently homogeneous in terms of participants involved, interventions, and outcomes through a brief search by authors. Thus, we added, "to the best of our knowledge, there are sufficient studies publishes in the last two decades so the likelihood of a future meta-analysis is well-supported".
- P.7, L.227-231 - we rephrased and added: "a z-score of ±3.29 cut off 0.1% of scores with 99.9% of z-scores lying between -3.29 and +3.29, the threshold of normal distribution in this protocol".
- P. 8, L.289-293 - The phrasing was adjusted to include statements rather than questions.
Reviewer 2 Report
I would like to congratulate all of you for this study protocol. PA is a multidimensional concept that still needs clarification and theoretical background. I am looking forward to read the results of your study.
it was a really well-written paper with a clear and easy-to-read text. It was also very well supported by the literature. PA is a multidimensional concept. Many things (e.g. weight, height, family, school, community facilities, etc) have to be taken into consideration. So, a theoretical framework is essential in order to explain all the factors, or most of them, that affect PA. Especially now in the covid-19 era, PA intervention program designs must motivate children and adolescents to participate in physical activities. The topic itself isn't so innovative. Many researchers try to study PA and the relative factors. But their design (systematic review and meta-analysis) wants to interpret all relevant factors that affect PA within the ecological theory, is promising. The questions addressed were not answered because the study will be conducted. So, results and conclusions were only assumed. This study protocol, in my opinion, can be an example of how you compose systematic literature review and meta-analysis.
Author Response
Thank you for your kind words and support! We appreciate the specific comments and the justification for the protocol and subsequent study.
Reviewer 3 Report
The manuscript entitled: "Physical Activity Participation and the Environment in Children and Adolescent: A Systematic Review and Meta-Analysis Protocol". This protocol plans an interesting systematic review. However, this protocol lacks crucial information that would be indispensable in any systematic review protocol. I hope my comments/suggestions/doubts help authors address some of these inconsistencies and improve their manuscript.
Line 81: It would be good that authors clarify what they mean by youth. Depending on the reference, youth might include young adults and not necessarily adolescents or younger children.
Line 118: Google scholar is not a scientific database. It is a search engine. This should be amended.
Table 1: I am wondering if authors will include Physical Education sessions as part of PA. If so, it would be helpful to add this word to the search strategy. If not, a rationale on why this is being excluded.
Line 133: does the age selection criteria will be considered at the involvement of the study? Some longitudinal studies might include people <18 years; however, they might turn>18 years as they are followed up. Are these types of studies going to be included/excluded?
It is unclear from the protocol if only intervention studies will be included. The authors mention that the designs include "longitudinal and cross-sectional" Unclear if this refers to observational studies. Later in 2.2.3 they seem that only intervention studies will be included? Unclear, please clarify.
Regarding assessing Bias' risk, the authors mention "The Cochrane Collaboration's tools"; however, this is quite broad and inaccurate. The refs 47 and 48 are not linked to the Cochrane collaboration tool (or any Risk of Bias assessment tool). This is an extremely sensitive issue in the protocol, considering that the types and studies included are broad and varied. Authors need to clarify what tools they will use and how the risk of bias/quality of each type and design of studies will be assessed.
Also, considering that authors will not only include randomised trials, it is unclear how the GRADE system will be implemented in such a heterogeneous sample of studies.
It is not clear from the protocol how the Risk of Bias will be included in the analysis.
The data synthesis only considers quantitative data. However, it is confusing since authors are also planning to include "mixed-methods" papers. Please clarify. Also, if a paper meets the inclusion criteria, but does not report results in the suggested variables, is there any plan for narrative review? If so, clarify.
The Heterogeneity and publication bias sections are only considering those studies that will be included in a meta-analysis. However, the authors mention that "If it is possible to conduct a meta-analysis". So, if there is no meta-analysis, what will happen with this analysis? Especially with publication bias?
General comments
References should be revised… References are missing, and some numbers do not correspond to what is being described in the text.
Add the definition of the acronyms the first time this appears in the text. E.g., line 31 PA.
Line 235: should it be "the researchers"?
Author Response
Thank you for your thoughtful feedback. Please see below our responses:
- P.1, L.19 - We removed Google scholar as a bibliographic database in this study. The remainder of the document was reviewed to ensure consistency in the manuscript, p.3, line 120 and Table 1 were revised accordingly.
- P.1, L.31 - We added the introduction of the abbreviation for physical activity (PA) in both the abstract and in the introduction on p. 1. The remainder of the document was reviewed to ensure other abbreviations were introduced prior to being used. The introduction of abbreviations for light physical activity, moderate physical activity, and moderate to vigorous physical activity were removed because the abbreviations were not subsequently used.
- P.3, L.82-83 - We added, "which referred to school-age (elementary, middle, and secondary) children and adolescents" to further explain youth in the context. We cited reference [36] because, in Welk (1999), they directly used the term "youth" and their definition is in line with this protocol. Later, in the method/participants section, the inclusion criteria for the age group is clear for the planned SR/MA. Meanwhile, we removed reference [38] because it discussed the ecological models but it was not limited to children and adolescents.
- Table 1 - We added Physical Education session(s) in Table 1 search strategy considered as one of the influences factors.
- P.4, L.135-136 - We added criteria for mixed-method studies. Only quantitative data in a mixed-method study will be included based on the inclusion criteria. In addition, qualitative research or data will not be reviewed because it is not our primary interest, and there is a paper reviewed qualitative studies in the relevant research area (please see the following citation). Citation: Allender, S., Cowburn, G., & Foster, C. Understanding participation in sport and physical activity among children and adults: a review of qualitative studies. Health Education Research. 2006, 21, 826-835.
- P.4, L.144-148 - We revised and added information to explain if the longitudinal design is included. More specifically, we will determine participants’ biological age as registered for the research at the start of the study. If participant’s age at the pre-test meets the 3-18-years-old criteria, this study is identified as meeting the participation inclusion criteria even though they might turn >18 years. Our justification is that those types of studies addressed adolescent PA, and also, include longitudinal data to reflect chronosystem factors in sociohistorical contexts.
- P.5, L.150 - We removed “PA intervention” in line 150 because PA interventions are under the health promotion interventions’ umbrella which mentioned in Table 1, in influence factors. It will be redundant to include both influence factors and PA interventions, so we removed it from 2.2.3. Further, this protocol will not be limited to intervention studies. More specially, RCT and observational studies (cohort studies, case-control studies, and cross-sectional studies) will all be included.
- P.5, L.181-185 - If a paper meets the inclusion criteria, but does not report results in the suggested variables, we will remove this study during the validation process. Because we addressed the criteria in p.5, lines 167-170 that outcomes should include PA outcome variables. Therefore, we added further clarification in the manuscript.
- P.6, L.198-203 - We addressed the “The Cochrane Collaboration's tools” adjustments as follows. We will use Cochrane Collaboration’s tool for assessing risk of bias in randomized trials. The updated citation to replace both previous citations [47, 48]: Higgins, J. P. T., Altman, D. G., Gøtzsche, P. C., Jüni, P., Moher, D., Oxman, A. D., Savovic, J., Schulz, K. F., Weeks, L., Sterne, J. A. C., & Cochrane Bias Methods Group. The Cochrane Collaboration's tool for assessing risk of bias in randomised trials. BMJ (Clinical research ed.). 2011, 343, d5928. [d5928]. https://doi.org/10.1136/bmj.d5928 In the meantime, we removed the previous citation [48] which is Campbell and Stanley (1963). Related citations are updated and reordered.
- P.6, L.203-209 - We will keep the plan to use the GRADE system, however, the risk of bias in non-randomized studies of interventions (ROBINS-I) tool as part of GRADE’s certainty rating process is going to be used for observational/non-randomized trials. In this way, the use of GRADE and ROBINS-I can allow authors to assess risk of bias in both RCTs and non-randomized studies on a common metric. The newly added citation for ROBINS-I: Schünemann, H. J., Cuello, C., Akl, E. A., Mustafa, R. A., Meerpohl, J. J., Thayer, K., Morgan, R.L., Gartlehner, G., Kunz, R., Katikireddi S.V., Sterne, J., Higgins, J.P., Guyatt, G., & GRADE Working Group. GRADE guidelines: 18. How ROBINS-I and other tools to assess risk of bias in nonrandomized studies should be used to rate the certainty of a body of evidence. J. Clin. Epidemiol. 2019, 111, 105-114. https://doi.org/10.1016/j.jclinepi.2018.01.012 Besides, we revised in-text citations ( p.7, Line 231; Line 236; Line 240; Lines 249-250; Lines 270; p.8, Line 278) in remainder document to be consistency.
- P.6, L.221-223 - The meta-analysis will be conducted if adequate studies are identified. To the best of our knowledge, there are sufficient studies published in the last two decades, so the likelihood of a future meta-analysis is well-supported. Thus, we have confidence that meta-analysis is needed in this study. We added further clarification in the manuscript.
- P.7, L.269 - Revised to "The researchers"
- p.10-11, L.420-441 - We revised and added new references, and in-text citation numbers were revised to match their associated content.
Round 2
Reviewer 3 Report
The authors have addressed all the issues raised. This is a better version of the protocol.